# Urinary Potassium Excretion, Fibroblast Growth Factor 23, and Incident Hypertension in the General Population-Based PREVEND Cohort

**DOI:** 10.3390/nu13124532

**Published:** 2021-12-17

**Authors:** Stanley M. H. Yeung, Ewout J. Hoorn, Joris I. Rotmans, Ron T. Gansevoort, Stephan J. L. Bakker, Liffert Vogt, Martin H. de Borst

**Affiliations:** 1Department of Internal Medicine, Division of Nephrology, University Medical Center Groningen, Hanzeplein 1, 9700 RB Groningen, The Netherlands; r.t.gansevoort@umcg.nl (R.T.G.); s.j.l.bakker@umcg.nl (S.J.L.B.); m.h.de.borst@umcg.nl (M.H.d.B.); 2Department of Internal Medicine, Division of Nephrology & Transplantation, Erasmus Medical Center, University Medical Center Rotterdam, 3015 GD Rotterdam, The Netherlands; e.j.hoorn@erasmusmc.nl; 3Department of Internal Medicine, Leiden University Medical Center, 2333 ZA Leiden, The Netherlands; j.i.rotmans@lumc.nl; 4Department of Internal Medicine, Section of Nephrology, Amsterdam Cardiovascular Sciences, Amsterdam University Medical Centers, University of Amsterdam, 1105 AZ Amsterdam, The Netherlands; l.vogt@amsterdamumc.nl

**Keywords:** diet, potassium, FGF23, hypertension, epidemiology

## Abstract

High plasma fibroblast growth factor 23 (FGF23) and low potassium intake have each been associated with incident hypertension. We recently demonstrated that potassium supplementation reduces FGF23 levels in pre-hypertensive individuals. The aim of the current study was to address whether 24-h urinary potassium excretion, reflecting dietary potassium intake, is associated with FGF23, and whether FGF23 mediates the association between urinary potassium excretion and incident hypertension in the general population. At baseline, 4194 community-dwelling individuals without hypertension were included. Mean urinary potassium excretion was 76 (23) mmol/24 h in men, and 64 (20) mmol/24 h in women. Plasma C-terminal FGF23 was 64.5 (54.2–77.8) RU/mL in men, and 70.3 (56.5–89.5) RU/mL in women. Urinary potassium excretion was inversely associated with FGF23, independent of age, sex, urinary sodium excretion, bone and mineral parameters, inflammation, and iron status (St. β −0.02, *p* < 0.05). The lowest sex-specific urinary potassium excretion tertile (HR 1.18 (95% CI 1.01–1.37)), and the highest sex-specific tertile of FGF23 (HR 1.17 (95% CI 1.01–1.37)) were each associated with incident hypertension, compared with the reference tertile. FGF23 did not mediate the association between urinary potassium excretion and incident hypertension. Increasing potassium intake, and reducing plasma FGF23 could be independent targets to reduce the risk of hypertension in the general population.

## 1. Introduction

Hypertension is a major modifiable risk factor for cardiovascular disease and premature death [1]. The global prevalence of hypertension is increasing, affecting >1.2 billion people [1]. Lifestyle interventions, including optimization of potassium intake, form an important strategy to prevent or control hypertension [2]. The World Health Organization endorses a minimal average intake of 90 mmol potassium every day; however, most individuals consuming a Western diet may not reach that amount [3,4].

Several observational studies have shown that low potassium intake is associated with hypertension [5,6,7]. In addition, randomized controlled trials found that potassium supplementation reduces blood pressure [8], probably at least in part through stimulation of natriuresis [9]. Interestingly, a number of animal and human studies have shown that higher potassium intake may influence plasma phosphate and fibroblast growth factor 23 (FGF23) levels [10,11,12]. Recently, it was demonstrated that potassium supplementation reduces plasma FGF23 levels in pre-hypertensive individuals [12].

FGF23, a phosphaturic hormone mainly secreted by osteocytes, is strongly associated with the progression of kidney function decline, iron deficiency, and inflammation [13,14]. Observational studies have shown that increased levels of FGF23 are associated with incident hypertension in the general population [15,16]. FGF23 may contribute to RAAS activation, and promote renal sodium reabsorption driving the risk of hypertension [17,18].

Together, these discoveries indicate that variation in potassium intake may influence FGF23 levels, and that the beneficial effects on blood pressure by potassium intake might partly occur via the lowering of FGF23. However, it has not yet been assessed whether urinary potassium excretion, as proxy for dietary potassium intake, is inversely associated with FGF23, and whether FGF23 mediates the association between urinary potassium excretion and incident hypertension.

This study adds three novel aspects to the literature. First, the association between urinary potassium excretion and FGF23 was examined, independent of potential confounders. Second, the associations between both baseline urinary potassium excretion and FGF23 and incident hypertension were assessed. Lastly, this study evaluated the potential mediation effect of FGF23 for the association between urinary potassium excretion and incident hypertension.

## 2. Materials and Methods

### 2.1. Study Design and Population

The present study was executed using data of Prevention of Renal and Vascular End-Stage Disease (PREVEND) study, which is a prospective cohort that aimed to study the natural course of elevated urinary albumin excretion levels in relationship with cardiovascular and kidney outcomes. In short, all individuals living in Groningen, The Netherlands, who were 28–75 years old, received a questionnaire, and were asked to collect a first-morning void urine between 1997 and 1998. Participants with a urinary albumin concentration (UAC) ≥ 10 mg/L could participate in the cohort (*n* = 7768). A second group, which was randomly selected from the population with a UAC < 10 mg/L, was asked to participate (*n* = 3394). Finally, a total of 8592 individuals were enrolled. A second study round took place between 2001 and 2003 in 6894 participants; this is considered the baseline of the present study. For current analyses, 4194 participants who were non-hypertensive at baseline and had available urinary potassium excretion and plasma FGF23 data were included. The University Medical Center Groningen Medical Ethics Committee has provided approval for the PREVEND study (ethical approval code: METC 96/01/022), which has been performed in accordance with Declaration of Helsinki guidelines. All participants provided written informed consent.

### 2.2. Data Collection

Data collection took place during two outpatient clinic visits separated by three weeks. Information on demographics, cardiovascular history, smoking habits, alcohol use, and medication use was obtained by questionnaires. Medication data were additionally obtained from IADB.nl, which is a database that contains information from all pharmacies in the city of Groningen. In the last week before the second visit, participants collected, in accordance to oral and written instructions, two consecutive 24 h urine samples. Urine was kept at 4 °C for up to four days before being stored at −20 °C. The average of both visits during baseline examination was used to calculate 24-h urinary potassium excretion. Furthermore, fasting blood samples were obtained and stored directly at −80 °C. Blood pressure was measured using an automatic device (Dinamap XL Model 9300 series, Johnson-Johnson Medical, Tampa, FL) on the right arm in supine position, every minute for 10 and 8 min, respectively. The average of the last two recordings was used. Body mass index (BMI) was calculated using weight (kilograms) divided by height squared (square meter). Smoking status was categorized as current smoker, former smoker, or never smoker. Alcohol intake was categorized as current or no alcohol consumption.

### 2.3. Study Outcome

The definition of hypertension was systolic blood pressure of ≥140 mmHg, diastolic blood pressure ≥90 mmHg, or the use of antihypertensive drugs. The primary outcome was incident hypertension, defined as the development of hypertension after baseline.

### 2.4. Laboratory Measurements

Potassium, sodium, and creatinine in the urine samples were determined with a MEGA clinical chemistry analyzer (Merck, Darmstadt, Germany). Fractional sodium excretion was calculated by ((urininary sodium × plasma creatinine)/(plasma sodium × urinary creatinine)). Plasma potassium, and sodium were determined by indirect potentiometry. Urinary albumin concentration was determined by nephelometry (Dade Behring Diagnostic, Marburg, Germany). Serum and urinary creatinine were determined using Kodak Ektachem dry chemistry (Eastman Kodak, Rochester, NY, USA). C-reactive protein was measured by nephelometry (BN II, Dade Behring, Marburg, Germany). Plasma 25-hydroxy (OH) vitamin D3 levels were measured using liquid chromatography tandem mass spectrometry, with intra- and inter-assay coefficients of variation for 25(OH)D of 7.2% and 6.7%, respectively. Plasma intact parathyroid hormone (PTH) concentrations were measured using an automated two-site immunoassay (Roche, Diagnostics, Indianapolis, IN, USA) with an intra-assay coefficient of variation 3.4% to 5.8%. Plasma circulating calcium and phosphate were determined in lithium heparin. Corrected calcium (mmol/L) was calculated as measured calcium (mmol/L) + 0.02 × (40 − serum albumin). Serum iron was measured using a colorimetric assay, ferritin using immunoassay, and transferrin using an immunoturbidimetric assay (all Roche Diagnostics). Transferrin saturation (TSAT, %) was calculated as 100 × serum iron (μmol/L) ÷ 25 × transferrin (g/L) [19]. Serum EPO was measured using an immunoassay based on chemiluminescence (Immulite EPO assay, DPC, Los Angeles, CA, USA). Serum hepcidin was measured with a competitive enzyme-linked immunosorbent assay (ELISA), as described elsewhere with intra- and inter-assay coefficients of variation (CVs) of 8.6% and 16.2%, respectively [20]. Estimated glomerular filtration rate (eGFR) was calculated based on serum creatinine Chronic Kidney Disease Epidemiology Collaboration (CKD-EPI) equation [21]. Urinary albumin-to-creatinine ratio (ACR) was calculated by dividing urinary albumin concentration (mg/L) by urinary creatinine concentration (mmol/L). C-terminal FGF23 (cFGF23) was measured in plasma by an enzyme-linked immunosorbent assay (ELISA) kit (Quidel, San Diego, CA, USA). This ELISA, which uses two antibodies targeting different epitopes within the C-terminal branch of FGF23, has an intra-assay coefficient of variation of <5%, and an inter-assay coefficient of variation of <16% in blinded replicated samples.

### 2.5. Statistical Analysis

Baseline characteristics are shown according to sex-specific tertiles of urinary potassium excretion and FGF23. Continuous data are given as mean, standard deviation (SD), or as median and interquartile range (IQR) in case of non-normal distribution. Categorical data are provided as percentiles. Variables with a skewed distribution were logarithmically transformed before use in subsequent analyses, where appropriate.

In order to avoid potential bias from the exclusion of patients with missing values, multiple imputation (fully conditional specification (Markov chain Monte Carlo)) was used to obtain five imputed datasets [22]. Rubin’s rules were used to obtain pooled estimates of the regression coefficients and their standard errors across imputed datasets [23].

Multivariable linear regression was used to analyze whether urinary potassium excretion was independently associated with FGF23. FGF23 was log_2_ transformed to obtain the best model fit. Model 1 was adjusted for age, sex, BMI, eGFR and urinary sodium, and phosphate excretion. In subsequent models, we performed additional adjustments for variables which are associated with FGF23 according to the literature. Model 2 was adjusted for bone mineral parameters (PTH, vitamin D, plasma phosphate, and calcium). Model 3 was adjusted for inflammation (CRP). We further adjusted for iron status. Model 4 was adjusted for ferritin, and model 5 for TSAT. Potential interaction effects by age, sex, and BMI were assessed by fitting models containing both main effects and their cross-product terms. Furthermore, it was assessed whether urinary potassium excretion and FGF23 are associated with systolic and diastolic blood pressure at baseline using multivariable linear regression analyses. Models were adjusted for age, BMI, eGFR, urinary sodium excretion, urinary albumin-to-creatinine ratio, smoking status, alcohol intake, diabetes, and cardiovascular disease.

Thereafter, Cox proportional hazard regression was used to examine the association of both urinary potassium excretion and FGF23 with incident hypertension. We cumulatively adjusted for confounders to determine independent association of urinary potassium excretion and FGF23. The following general models were built: model 1 was adjusted for age BMI, eGFR, sodium intake, estimated as urinary sodium excretion, and urinary albumin-to-creatinine ratio. In addition, model 2 was adjusted for important risk factors for risk for hypertension, including smoking status (current, former, or never), alcohol intake, diabetes, and cardiovascular disease. In model 3, the association between urinary potassium excretion and incident hypertension was adjusted for FGF23 to find a potential mediation effect of FGF23. Cox regression analyses of FGF23 and incident hypertension were additionally adjusted for bone and mineral parameters (PTH; vitamin D; phosphate intake, estimated as urinary phosphate excretion; plasma corrected calcium; and plasma phosphate) in model 3. The associations of urinary potassium excretion and FGF23 on incident hypertension are visualized by fitting multivariable Cox regression analyses according to model 2.

Statistical analyses were executed with SPSS software, version 23.0 for Windows (IBM, Armonk, NY, USA), and R version 4.0.1 (R Foundation for Statistical Computing, Vienna, Austria) (http://cran.r-project.org/, accessed on 30 June 2020). A 2-sided *p*-value < 0.05 was considered significant in all analyses.

## 3. Results

### 3.1. Baseline Characteristics

Mean urinary potassium excretion was 76 (23) mmol/24 h in men, and 64 (20) mmol/24 h in women. FGF23 levels were 64.5 (54.2–77.8) RU/mL in men, and 70.3 (56.5–89.5) RU/mL in women. Baseline characteristics of urinary potassium excretion and FGF23 according to sex-specific tertiles are shown in Table 1 and in Appendix A, respectively. Participants in the highest sex-specific tertile were younger, more likely to be overweight, had lower CRP, higher vitamin D, lower FGF23, higher urinary sodium, higher phosphate excretion, and higher TMP/GFR, compared with participants in the lowest sex-specific tertile (Table 1). Participants with higher plasma FGF23 were older, more likely to be overweight, had a lower eGFR, higher CRP, higher PTH and vitamin D, were more often iron deficient, and had lower urinary potassium excretion, and higher urinary ACR, compared with participants with lower FGF23 levels.

### 3.2. Urinary Potassium Excretion and FGF23

In multiple linear regression analyses, urinary potassium excretion was inversely associated with FGF23 levels in a multivariable analysis adjusted for age, sex, BMI, eGFR, urinary sodium excretion, and urinary phosphate excretion (Table 2). Further adjustment for bone and mineral parameters, inflammation, or iron status did not materially change the result (Table 2). No significant interaction by age, sex, or BMI was found (all P-interaction >0.1). Furthermore, in multivariable linear regression, neither urinary potassium excretion nor FGF23 were associated with baseline systolic or diastolic blood pressure (Appendix A).

### 3.3. Urinary Potassium Excretion, FGF23, and Incident Hypertension

In the course of a median follow-up for 7.1 (IQR 3.6–7.6) years, 1073 (29%) participants developed new-onset hypertension, whereas 474 participants were lost to follow-up. The lowest sex-specific tertile of urinary potassium excretion was associated with a higher risk of hypertension, compared with the second tertile (hazard ratio (HR) 1.20, (95% confidence interval (CI) 1.03–1.39), *p* = 0.02). This association persisted upon adjustment for potential confounders in model 2 (HR 1.18 (95% CI 1.01–1.37), *p* = 0.04) (Table 3). The adjusted association between urinary potassium excretion and incident hypertension is visualized in Figure 1. No potential mediation effect by FGF23 was found, as the point estimate of the HR of the association of urinary potassium excretion with incident hypertension did not change after adjustment for FGF23 (model 3, Table 3).

When using the lowest sex-specific tertile of FGF23 as the reference group, we observed that the highest sex-specific tertile of FGF23 was associated with a higher risk for incident hypertension (HR 1.27 (95% CI 1.07–1.51), *p* = 0.02). This association remained, independent of adjustment for potential confounders (model 3: HR 1.25 (95% CI 1.05–1.49), *p* = 0.04) (Table 4). The adjusted association between FGF23 and incident hypertension is depicted in Figure 2.

## 4. Discussion

In this large general population cohort, baseline urinary potassium excretion was inversely associated with plasma FGF23. Furthermore, low urinary potassium excretion, and high FGF23 levels were each associated with a higher risk of incident hypertension. However, no mediation effect of FGF23 was found between the inverse association of urinary potassium excretion and incident hypertension.

The current results are in line with findings from a prior observational dietary study comparing FGF23 levels in individuals consuming a Western diet (low potassium) versus a non-Western diet (high potassium) [24]. However, the results of this prior observational study might have been confounded by a concomitantly low phosphate intake in the non-Western diet [24,25]. Yet, the association between urinary potassium excretion and FGF23 was independent of urinary phosphate excretion, as proxy for dietary phosphate intake [26]. Furthermore, current results were in line with a short-term placebo and diet-controlled intervention trial wherein potassium chloride supplementation led to a decrease of FGF23 levels in pre-hypertensive individuals [12]. Similar to the interventional study, participants with higher urinary potassium excretion had a lower fractional phosphate excretion, and a higher TMP/GFR, which are expected findings if FGF23 is directly decreased by potassium [27]. In contrast, plasma phosphate concentrations did not differ between the urinary potassium excretion sex-specific tertiles, whereas in an intervention study, it was found that plasma phosphate increased after potassium supplementation [12]. This might indicate that the increase of plasma phosphate due to the lowering of FGF23 levels is a short-term effect. Higher vitamin D was observed in participants with higher urinary potassium excretion, pointing towards a healthier lifestyle, e.g., more sun exposure or a healthier diet in these participants [28]. Another potential explanation is that FGF23 suppresses vitamin D, and, therefore, participants with a high FGF23 have lower vitamin D [27]. It should be noted that the inverse association between urinary potassium excretion and FGF23 remained independent of FGF23-stimulating factors, such as inflammation and iron status [14,29]. Therefore, potential mediation by inflammation or iron status seems unlikely. It may be speculated that high potassium intake directly influences FGF23 production in osteocytes. In this respect, in vitro studies using bone cells revealed that a decrease in potassium concentration could stimulate bone resorption [30] which, in turn, might increase FGF23 production [31]. It has also been shown that potassium supplementation inhibits the sodium-chloride co-transporter (NCC) in the distal convoluted tubule [32]. Some studies showed that NCC is also expressed in osteoblasts, and that inhibition of NCC on osteoblasts might decrease FGF23 production [33,34].

In this subset from the PREVEND cohort without hypertension at baseline, it was found that both low potassium excretion and high FGF23 were associated with incident hypertension, in line with previously published studies [5,6,7,15,16]. Current results extend a prior analysis in the PREVEND cohort on the relationship between potassium excretion and incident hypertension by specifically addressing potential mediation by FGF23 [5]. It has been shown that low potassium intake most likely increases renal sodium reabsorption, leading to higher blood pressure [35]. Potassium supplementation was able to lower blood pressure [8], and this effect is partly attributed to a natriuretic effect, which is enhanced by the inhibition of NCC in distal kidney tubules [32]. Moreover, a recent large cluster randomized trial showed that salt substation by potassium salt reduces blood pressure, and lowers the risk of stroke and mortality in participants with a history of stroke [36]. Furthermore, potassium supplementation may improve vascular and endothelial function, and it may reduce vascular calcification, leading to lower blood pressure [37,38,39].

It has been suggested that FGF23 is not only a marker for progression of kidney function decline, but it might be a detrimental cardiovascular factor in itself. For example, it has been shown that FGF23 might increase the renin–aldosterone–angiotensin system by a decrease of active vitamin D synthesis [17]. It has also been suggested that FGF23 could inhibit angiotensin-converting enzymes. Moreover, FGF23 has been associated with sodium reabsorption, leading to higher blood pressure [18].

This study addressed whether FGF23 is a mediator of the association between urinary potassium excretion and incident hypertension. However, adding FGF23 to the multivariable model did not change the point estimate of the hazard ratio of the association between urinary potassium and incident hypertension. This is likely due to the small lowering effect of potassium intake on FGF23, which was of a similar magnitude as in the potassium supplementation interventional trial [12]. It was shown that other dietary interventions, such as low protein, low phosphate, low caloric intake, may decrease FGF23 levels [40,41,42]. A dietary pattern designed to lower FGF23, including an increase in potassium intake, may elicit a greater lowering effect of FGF23, with a subsequent risk reduction of cardiovascular outcome. Patients with CKD and kidney failure display higher FGF23 levels due to the strong association between FGF23 and kidney function [43]. It is not known whether potassium supplementation may decrease high FGF23 levels, and whether the effect size is different compared to pre-hypertensive individuals [12]. Currently, a double-blind randomized placebo trial is being conducted wherein potassium is supplemented to patients with a CKD grade 3b/4 to assess whether higher potassium intake hampers kidney function decline [44]. In addition, this study may reveal whether an increase in potassium intake may lower blood pressure in patients who are already using anti-hypertensive drugs, such as diuretics and ACE-inhibitors(angiotensin converting enzyme inhibitors or angiotensin receptor blockers).

This study was performed in a large and well-characterized cohort. Although no repeated 24-h urinary measurements were available during follow-up [45], two independent 24-h urine collections at baseline were used to estimate potassium intake. No renin data are available, which precludes analyses that address possible interactions between renin, urinary potassium excretion, and FGF23. Findings from the PREVEND cohort, which consisted predominantly of individuals of European ancestry, may not be generalizable to other populations [46]. Finally, these findings are observational, and, therefore, residual confounding might persist despite multivariable adjustment.

## 5. Conclusions

Urinary potassium excretion, as proxy for dietary potassium intake, was inversely associated with FGF23 in the general population even after adjustment for FGF23-stimulating factors. Furthermore, low urinary potassium excretion and high FGF23 are independently associated with incident hypertension, but FGF23 was not a mediating factor between urinary potassium excretion and incident hypertension.

## Figures and Tables

**Figure 1 nutrients-13-04532-f001:**
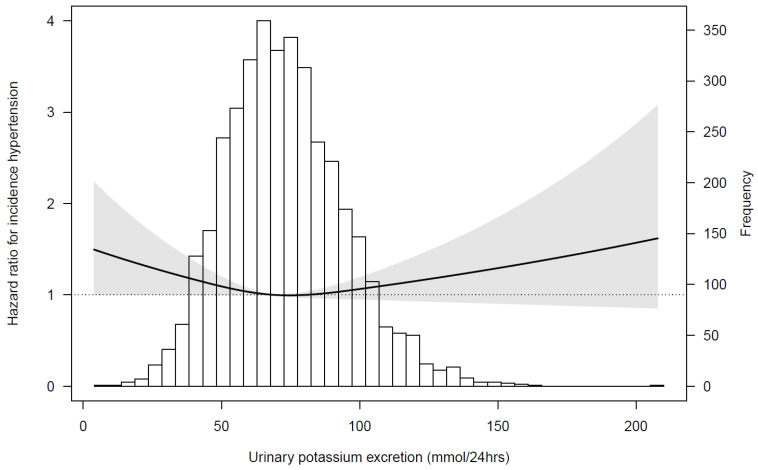
Associations between urinary potassium excretion and incident hypertension in 3720 participants. Data were fit by a Cox proportional hazards regression model adjusted for age, BMI, eGFR, urinary sodium excretion, urinary albumin-to-creatinine ratio, smoking status, alcohol consumption, diabetes, history of cardiovascular disease, anti-diabetic, and lipid lowering drugs. The grey area represents the 95% CI. Abbreviations: eGFR, estimated glomerular filtration rate; BMI, body mass index.

**Figure 2 nutrients-13-04532-f002:**
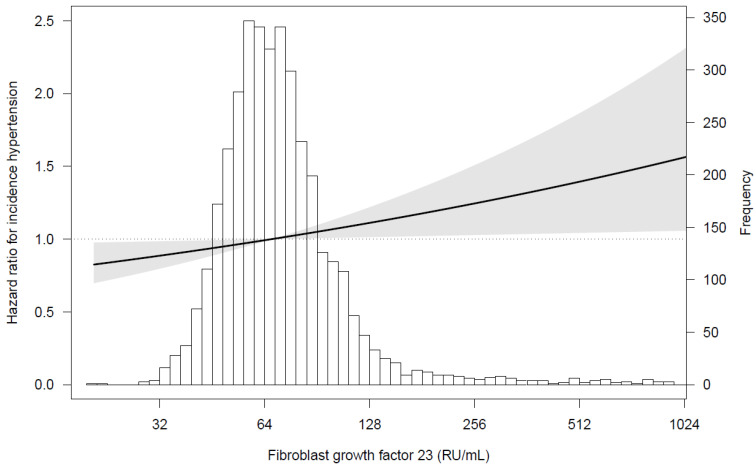
Associations between fibroblast growth factor 23 and incident hypertension in 3720 participants. Data were fit by a Cox proportional hazards regression model adjusted for age, BMI, eGFR, urinary sodium excretion, urinary albumin-to-creatinine ratio, diabetes, smoking, history of cardiovascular disease, anti-diabetic, lipid lowering drugs, PTH, vitamin D, plasma phosphate, plasma corrected calcium, and urinary phosphate excretion. The grey area represents the 95% CI. Abbreviations: eGFR, estimated glomerular filtration rate; BMI, body mass index; PTH, parathyroid hormone.

**Table 1 nutrients-13-04532-t001:** Baseline parameters according to sex-specific tertiles of urinary potassium excretion in 4194 participants of PREVEND.

	Sex-Specific Tertiles of Total Urinary Potassium Excretion, mmol/24-h			
	I	II	III	*p*-Trend *
Men	<66	66–85	>85	
Women	<55	55–72	>72	
Urinary potassium excretion, (mmol/24 h)	47 (10)	68 (8)	93 (16)	
Men, *n* (%)	647 (46)	648 (47)	648 (46)	
Age, (year)	50 (11)	50 (10)	49 (10)	<0.001
BMI, (kg/m^2^)	25.0 (22.8–27.6)	25.2 (23.0–27.8)	25.5 (23.5–28.3)	<0.001
Systolic blood pressure, (mmHg)	117 (109–126)	117 (109–127)	117 (110–127)	0.36
Diastolic blood pressure, (mmHg)	70 (65–76)	70 (65–75)	70 (65–76)	0.70
Lipid lowering drugs, *n* (%)	53 (4)	39 (3)	37 (3)	0.24
Antidiabetic drugs, *n* (%)	15 (1)	19 (2)	20 (2)	0.58
Smoking status				0.007
Never, *n* (%)	386 (28)	449 (32)	455 (33)	
Former or current, *n* (%)	1008 (72)	942 (68)	938 (67)	
Alcohol consumption, yes (%)	999 (72)	1095 (78)	1140 (82)	<0.001
Cardiovascular disease, yes (%)	55 (5)	56 (6)	72 (8)	0.15
Diabetes, yes (%)	43 (3)	39 (3)	36 (3)	0.74
eGFR (CKD-epi), (mL/min·1.73 m^2^)	97 (14)	97 (13)	97 (13)	0.33
Plasma albumin, (g/L)	44 (4)	44 (5)	44 (6)	0.43
Plasma potassium, (mmol/L)	4.2 (0.3)	4.2 (0.3)	4.2 (0.2)	<0.001
Plasma sodium, (mmol/L)	141 (2)	141 (2)	141 (2)	0.17
High sensitive CRP, (mg/L)	1.23 (0.59–2.85)	1.07 (0.50–2.47)	1.02 (0.48–2.22)	<0.001
Plasma phosphate, (mmol/L)	1.02 (0.25)	1.05 (0.53)	1.02 (0.32)	0.07
Plasma PTH, (pmol/L)	4.8 (4.0–5.8)	4.7 (3.9–5.7)	4.7 (3.9–5.5)	0.08
Plasma vitamin D_3_, 25-OH, (nmol/L)	55.1 (38.7–74.9)	57.8 (40.8–77.2)	58.6 (42.8–77.7)	0.001
Plasma fibroblast growth factor 23 (RU/mL)	70 (56–87)	68 (56–84)	66 (55–81)	0.001
Plasma calcium, (mmol/L)	2.29 (0.12)	2.29 (0.12)	2.29 (0.10)	0.66
Plasma corrected calcium, (mmol/L)	2.22 (0.15)	2.21 (0.17)	2.21 (0.15)	0.68
Iron, (umol/L)	16 (6)	16 (6)	16 (6)	0.62
Ferritin, (μg/L)	86 (41–157)	79 (39–144)	79 (38–149)	0.29
Transferrin saturation, (%)	25 (10)	25 (9)	25 (9)	0.75
Urinary sodium excretion, (mmol/24 h)	112 (88–146)	139 (110–173)	160 (131–198)	<0.001
Urinary phosphate excretion, (mmol/24 h)	21 (17–27)	25 (20–32)	30 (23–37)	<0.001
TMP/GFR	0.97 (0.80–1.18)	1.01 (0.82–1.22)	1.03 (0.83–1.24)	<0.001
Urinary creatinine excretion, (mmol/24 h)	11 (3)	12 (3)	14 (3)	<0.001
Urinary albumin excretion, (mg/24 h)	10.5 (7.8–16.7)	11.7 (8.7–17.3)	12.5 (9.4–20.6)	<0.001
Urinary albumin-to creatinine ratio, (mg/mmol)	0.66 (0.48–1.06)	0.64 (0.48–0.99)	0.64 (0.48–1.04)	0.70

Data are showed as *n* (%), mean (SD), or median (interquartile range) for nominal, normal distributed, and non-normal distributed data, respectively. * The *p*-value represents the p for trend using Chi-squared test, one-way-ANOVA, or Kruskal–Wallis test for nominal, normal distributed, and non-homogeneity or non-normal distributed data, respectively. Abbreviations: BMI, body mass index; eGFR (CKD-epi), estimated glomerular filtration rate based on Chronic Kidney Disease Epidemiology Collaboration equation; CRP, C-reactive protein; PTH, parathyroid hormone.

**Table 2 nutrients-13-04532-t002:** Multivariable associations between urinary potassium excretion and (log_2_) fibroblast growth factor 23 in 4194 participants of the PREVEND cohort.

	Difference per Unit Standardized Variable in Urinary Potassium Excretion with FGF23	
Model	Standardized Beta	*p*-Value
1	−0.04	0.02
2	−0.04	0.05
3	−0.04	0.02
4	−0.06	0.002
5	−0.04	0.02

Model 1: adjusted for age, sex, BMI, eGFR, urinary sodium excretion, urinary phosphate excretion. Model 2: as model 1, and additionally adjusted for ln (PTH), ln (vitamin D), plasma phosphate, and plasma corrected calcium. Model 3: as model 1, and additionally adjusted for CRP. Model 4: as model 1, and additionally adjusted for ferritin. Model 5: as model 1, and additionally adjusted for TSAT.

**Table 3 nutrients-13-04532-t003:** Associations between urinary potassium excretion and risk of incident hypertension in 3720 PREVEND participants.

	Sex-Specific Tertiles of Urinary Potassium Excretion Hazard Ratio (95% CI)		
	I	II	III
Men	<66 mmol/24 h	66–85 mmol/24 h	>85 mmol/24 h
Women	<55 mmol/24 h	55–72 mmol/24 h	>72 mmol/24 h
Model 1	1.20 (1.03–1.39) *	1.0 (ref.)	1.08 (0.93–1.26)
Model 2	1.19 (1.02–1.38) *	1.0 (ref.)	1.09 (0.94–1.26)
Model 3	1.19 (1.02–1.38) *	1.0 (ref.)	1.09 (0.94–1.26)

Model 1: adjusted for age, BMI, eGFR, urinary sodium excretion, and urinary albumin-to-creatinine ratio. Model 2: as model 1, and additionally adjusted for smoking status, alcohol consumption, diabetes, and history of cardiovascular disease. Model 3: as model 2, and additionally adjusted for fibroblast growth factor 23. Data are presented as HR, hazard ratio; 95% CI, confidence interval; *p*-value is shown as: * ≤ 0.05. Abbreviations: CI, confidence interval; BMI, body mass index; eGFR, estimated glomerular filtration rate.

**Table 4 nutrients-13-04532-t004:** Associations between fibroblast growth factor 23 and risk of incident hypertension in 3720 PREVEND participants.

	Sex-Specific Tertiles of Fibroblast Growth Factor 23 Hazard Ratio (95% CI)		
	I	II	III
Men	<58 RU/mL	58–73 RU/mL	≥73 RU/mL
Women	<61 RU/mL	61–82 RU/mL	≥82 RU/mL
Model 1	1.0 (ref.)	1.02 (0.88–1.19)	1.19 (1.03–1.38) *
Model 2	1.0 (ref.)	1.00 (0.86–1.16)	1.17 (1.01–1.36) *
Model 3	1.0 (ref.)	1.00 (0.86–1.17)	1.17 (1.01–1.37) *

Model 1: adjusted for age, BMI, eGFR, urinary sodium excretion, and urinary albumin-to-creatinine ratio. Model 2: as model 1, and additionally adjusted for diabetes, smoking, and history of cardiovascular disease. Model 3: as model 2, and additionally adjusted for PTH, vitamin D, plasma phosphate, plasma corrected calcium, and urinary phosphate excretion. Data are presented as HR, hazard ratio; 95% CI, confidence interval; *p*-value is shown as: * ≤ 0.05. Abbreviations: CI, confidence interval; BMI, body mass index; eGFR, estimated glomerular filtration rate.

## Data Availability

The data underlying the results presented in this study can be made available by the data manager of the PREVEND study, Lyanne Kieneker, l.m.kieneker@umcg.nl. Public sharing of individual participant data was not included in the informed consent form of the study, but data can be made available to interested researchers upon reasonable request.

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
