# Peer review of "Urinary Potassium Excretion, Fibroblast Growth Factor 23, and Incident Hypertension in the General Population-Based PREVEND Cohort"

_nutrients, 2021, doi:10.3390/nu13124532_

Round 1

Reviewer 1 Report

The purpose of the study is to evaluate the relationship between potassium intake and FGF23. Good Results part. Please see my suggestions regarding the improvement of this paper.

References must be in brackets, not in parenthesis. Please check the Instructions for authors. https://www.mdpi.com/journal/nutrients/instructions.

English must be carefully checked and corrected in the entire manuscript. Please avoid using the personal manner of addressing “we”, “our”, (it is annoying so many “we”) and use the impersonal one; the text will sound much more professional (i.e., L58 "we assessed", L59 "We subsequently assessed" ; L338 and L340 "we found" - the text is too repetitive. Please revise accordingly.

Introduction part must be better developed. Please also highlight better the novelty that your research brings to the field in the aim of the study paragraph L58-62

The Material and methods of the study are adequately described and the statistical analysis is adjusted according to the purpose of the study. The subject is of great interest for the general population given the high incidence of hypertension and the failure of the public health measures to reduce hypertension prevalence.

The Discussions are relatively good but I suggest improving this part. Please comment on the situation when potassium is artificially corrected with ACEI drugs; is hyperpotassemia a reason  of ACEI everyday in treating high blood pressure? Is there any interaction between renin values, poatsium values and FGF 23? Please check: https://rjme.ro/RJME/resources/files/521111419423.pdf. Please discuss the possible impact of potassium supplementation in patients with high FGF23 values. Can, from public health point of view, potassium enrichment of certain foods lead to a decrease of hypertension prevalence? I suggest checking https://www.sciencedirect.com/science/article/pii/S0753332220309070. 

Author Response

The purpose of the study is to evaluate the relationship between potassium intake and FGF23. Good Results part. Please see my suggestions regarding the improvement of this paper.

Point 1: References must be in brackets, not in parenthesis. Please check the Instructions for authors. https://www.mdpi.com/journal/nutrients/instructions.

Response 1: Thank you for noting this. All references throughout the manuscript have been adapted, and are now in brackets.

Point 2: English must be carefully checked and corrected in the entire manuscript. Please avoid using the personal manner of addressing “we”, “our”, (it is annoying so many “we”) and use the impersonal one; the text will sound much more professional (i.e., L58 "we assessed", L59 "We subsequently assessed" ; L338 and L340 "we found" - the text is too repetitive. Please revise accordingly.

Response 2: Thank you for your suggestion. We have changed the manuscript accordingly , limiting the usage of “we” and “our”.

Point 3: Introduction part must be better developed. Please also highlight better the novelty that your research brings to the field in the aim of the study paragraph L58-62

Response 3: We have adjusted parts of the introduction and we have highlighted the novelty of our research at the end of the introduction.

The Material and methods of the study are adequately described and the statistical analysis is adjusted according to the purpose of the study. The subject is of great interest for the general population given the high incidence of hypertension and the failure of the public health measures to reduce hypertension prevalence.

Point 4: The Discussions are relatively good but I suggest improving this part. Please comment on the situation when potassium is artificially corrected with ACEI drugs; is hyperpotassemia a reason  of ACEI everyday in treating high blood pressure? Is there any interaction between renin values, poatsium values and FGF 23? Please check: https://rjme.ro/RJME/resources/files/521111419423.pdf. Please discuss the possible impact of potassium supplementation in patients with high FGF23 values. Can, from public health point of view, potassium enrichment of certain foods lead to a decrease of hypertension prevalence? I suggest checking https://www.sciencedirect.com/science/article/pii/S0753332220309070.

Response 4: Thank you for your interesting remarks. The blood pressure-lowering effect of ACEI are generally considered to be mostly driven by lowering of angiotensin II through inhibition of the angiotensin-converting enzyme. Since hyperkalemia is a relatively common side effect of ACEI, and higher potassium intake reduces blood pressure, it is conceivable that plasma potassium mediates in part the antihypertensive effect of ACEI. Unfortunately, the observational nature of the current study precludes the isolation of the (potential) potassium-driven effect of ACEI from the angiotensin II-driven effect.

Second, since no renin data are available, it is not possible to assess the interaction between renin values, potassium values and FGF23. We have added this to the limitations section of the revised manuscript.

Third, a previous study reported that potassium supplementation lowers FGF23 in individuals with normal kidney function [1]. A recent follow-up study revealed that potassium has similar effects on FGF23 in patients with higher FGF23 levels (i.e., patients with chronic kidney disease). Unfortunately, since these data have not yet been published, we cannot discuss this in detail in the current manuscript.

Fourth, we are currently undertaking a double-blind randomized placebo-controlled trial to investigate the effects of potassium supplementation on blood pressure and kidney function decline in patients with a chronic kidney disease stage 3b/4 [2]. The potential benefits of higher potassium intake were further highlighted by a recent large trial showing that salt substitution by potassium salt led to lower blood pressure, and lower risks of stroke and mortality [3]. We have added to the discussion that higher potassium intake might lead to lower blood pressure and reduce the risk of stroke and mortality [3].

References

  1. Humalda JK, Yeung SMH, Geleijnse JM, Gijsbers L, Riphagen IJ, Hoorn EJ, Rotmans JI, Vogt L, Navis G, Bakker SJL, et al. Effects of Potassium or Sodium Supplementation on Mineral Homeostasis: A Controlled Dietary Intervention Study. Vol. 105, The Journal of clinical endocrinology and metabolism. 2020 Sep 1;105(9):e3246–56.
  2. Gritter M, Vogt L, Yeung SMH, Wouda RD, Ramakers CRB, de Borst MH, Rotmans JI, Hoorn EJ. Rationale and Design of a Randomized Placebo-Controlled Clinical Trial Assessing the Renoprotective Effects of Potassium Supplementation in Chronic Kidney Disease. Vol. 140, Nephron. 2018;140(1):48–57.
  3. Neal B, Wu Y, Feng X, Zhang R, Zhang Y, Shi J, Zhang J, Tian M, Huang L, Li Z, et al. Effect of Salt Substitution on Cardiovascular Events and Death. Vol. 385, New England Journal of Medicine. 2021 Sep 16;385(12):1067–77.

Reviewer 2 Report

This is a well-written manuscript. The methods are clearly explained and the statistical approach is appropriate for answering the research question. The analysis of the potential impact of confounding variables adds to the significance of the findings.

Author Response

Thank you for your kind words and thank you for your time to review our manuscript.